# Dynamics of chromosomal target search by a membrane-integrated one-component receptor

**Linda Martini**[1], **Sophie Brameyer**[2], **Elisabeth Hoyer**[2], **Kirsten Jung**[2]*, **Ulrich Gerland**[1]*

**1** Physics of Complex Biosystems, Technical University of Munich, Garching, Germany, **2** Microbiology, Ludwig-Maximilians-University Munich, Martinsried, Germany

☯ These authors contributed equally to this work.

* jung@lmu.de (KJ); gerland@tum.de (UG)

**Data Availability Statement:** All relevant data are within the manuscript and its Supporting information files.

## Abstract

Membrane proteins account for about one third of the cellular proteome, but it is still unclear how dynamic they are and how they establish functional contacts with cytoplasmic interaction partners. Here, we consider a membrane-integrated one-component receptor that also acts as a transcriptional activator, and analyze how it kinetically locates its specific binding site on the genome. We focus on the case of CadC, the pH receptor of the acid stress response Cad system in *E. coli*. CadC is a prime example of a one-component signaling protein that directly binds to its cognate target site on the chromosome to regulate transcription. We combined fluorescence microscopy experiments, mathematical analysis, and kinetic Monte Carlo simulations to probe this target search process. Using fluorescently labeled CadC, we measured the time from activation of the receptor until successful binding to the DNA in single cells, exploiting that stable receptor-DNA complexes are visible as fluorescent spots. Our experimental data indicate that CadC is highly mobile in the membrane and finds its target by a 2D diffusion and capture mechanism. DNA mobility is constrained due to the overall chromosome organization, but a labeled DNA locus in the vicinity of the target site appears sufficiently mobile to randomly come close to the membrane. Relocation of the DNA target site to a distant position on the chromosome had almost no effect on the mean search time, which was between four and five minutes in either case. However, a mutant strain with two binding sites displayed a mean search time that was reduced by about a factor of two. This behavior is consistent with simulations of a coarse-grained lattice model for the coupled dynamics of DNA within a cell volume and proteins on its surface. The model also rationalizes the experimentally determined distribution of search times. Overall our findings reveal that DNA target search does not present a much bigger kinetic challenge for membrane-integrated proteins than for cytoplasmic proteins. More generally, diffusion and capture mechanisms may be sufficient for bacterial membrane proteins to establish functional contacts with cytoplasmic targets.

**Funding:** This work was supported by the German Research Council (DFG) within the framework of the Transregio 174 "Spatiotemporal dynamics of bacterial cells" (to K.J. and U.G.). The funders had no role in study design, data collection and analysis, decision to publish, or preparation of the manuscript.

**Competing interests:** The authors have declared that no competing interests exist.

## Author summary

Adaptation to changing environments is vital to bacteria and is enabled by sophisticated signal transduction systems. While signal transduction by two-component systems is well studied, the signal transduction of membrane-integrated one-component systems, where one protein performs both sensing and response regulation, are insufficiently understood. How can a membrane-integrated protein bind to specific sites on the genome to regulate transcription? Here, we study the kinetics of this process, which involves both protein diffusion within the membrane and conformational fluctuations of the genomic DNA. A well-suited model system for this question is CadC, the signaling protein of the *E. coli* Cad system involved in pH stress response. Fluorescently labeled CadC forms visible spots in single cells upon stable DNA-binding, marking the end of the protein-DNA search process. Moreover, the start of the search is triggered by a medium shift exposing cells to pH stress. We probe the underlying mechanism by varying the number and position of DNA target sites. We combine these experiments with mathematical analysis and kinetic Monte Carlo simulations of lattice models for the search process. Our results suggest that CadC diffusion in the membrane is pivotal for this search, while the DNA target site is just mobile enough to reach the membrane.

## Introduction

Bacteria are exposed to fluctuating environments with frequent changes in nutrient conditions and communication signals, but also life-threatening conditions such as environmental stresses and antibiotics [1]. To sense and adapt to changing environmental conditions, bacteria have evolved sophisticated signaling schemes, primarily based on one- and two-component systems [2, 3].

Two-component signaling systems feature a sensor kinase and a separate response regulator, where the former is typically membrane-integrated while the latter diffuses through the cytoplasm to reach its regulatory target [3]. The majority of response regulators are transcription factors that bind to specific target sites on the genomic DNA to activate or repress transcription. Hence, a key step in the signal transduction pathway of these two-component systems is a DNA target search by a cytoplasmic protein. The target search dynamics of cytoplasmic transcription factors have been thoroughly studied in the past decades, triggered by early *in vitro* experiments indicating that the *Escherichia coli* Lac repressor finds its target site faster than the rate limit for three-dimensional (3D) diffusion [4]. Inspired by the idea that a reduction of dimensionality can lead to enhanced reaction rates [5], the experiments were explained by a two-step process, where transcription factors locate their target by alternating periods of 3D diffusion and 1D sliding along the DNA [6]. Compared to pure 3D diffusion, sliding increases the association rate by effectively enlarging the target size ("antenna" effect) [7–9]. These dynamics were later probed with single-molecule methods, both *in vitro* [10, 11] and *in vivo* [12, 13]. For the Lac repressor as a paradigmatic example of a low copy number cytoplasmic transcription factor, the *in vivo* timescale of the target search was found to be about one minute [12]. Further studies continued to add to the detailed understanding of this target search process, e.g. with respect to effects of DNA conformation [14], DNA dynamics [15], and macromolecular crowding [16].

One-component signaling systems, in contrast, combine sensory function and response regulation within one protein [2]. The subset of one-component systems that are both membrane-integrated sensors and DNA-binding response regulators face an extraordinary DNA

target search problem: They must locate and bind to a specific site on the bacterial chromosome from the membrane. This is the case for one-component systems of the ToxR receptor family, which have a modular structure featuring a periplasmic sensory domain followed by a single transmembrane helix connected via a linker to a cytoplasmic DNA-binding domain [17]. In addition to ToxR, the main regulator for virulence in *Vibrio cholerae*, members of this receptor family include TcpP and TfoS in *V. cholerae* [18], PsaE in *Yersinia tuberculosis* [19] and the pH stress-sensing receptor CadC in *E. coli* [20]. *A priori*, it is not clear how these one-component systems establish functional protein-DNA contacts after stimulus perception. One scenario is that the DNA-binding domain is proteolytically cleaved, such that it can search for its specific binding site in the same manner as a regular cytoplasmic transcription factor. Another possible scenario is that simultaneous transcription, translation, and membrane-insertion ("transertion" [21, 22]) tethers the DNA locus of the one-component system to the membrane and, at the same time, places the protein in the vicinity of its binding site on the DNA (which is typically close to the gene encoding the one-component system). The third scenario is a diffusion and capture mechanism [23], whereby the one-component system diffuses within the membrane and captures conformational fluctuations that bring the DNA close to the membrane.

For the case of *E. coli* CadC, a well characterized model system [24], indirect evidence [25, 26] paired with direct observation [27] argues against the proteolytic cleavage and transertion scenarios, and instead supports the diffusion and capture mechanism. For instance, the transertion mechanism should be sensitive to relocating the *cadC* gene to a locus far from its native position, which is close to the CadC target site at the *cadBA* promoter. However, relocation to the distant *lac* operon did not reduce the regulatory output of CadC. One of the observations arguing against proteolytic cleavage was that external signals can rapidly deactivate the CadC response after the original stimulus, whereas cleavage would irreversibly separate the DNA-binding domain from the signaling input [25]. In contrast, the diffusion and capture mechanism appeared consistent with experiments that imaged fluorophore-labeled CadC *in vivo* [27]. These experiments showed that localized CadC spots in fluorescent microscopic images form after cells were shifted to a medium that simultaneously provides acid stress and a lysine-rich environment, the two input signals required to stimulate the CadC response [25]. Formation of these spots was rapidly reversible upon removal of the input signals. Furthermore, the observed number of CadC spots was positively correlated with the number of DNA-binding sites, indicating that the spots correspond to CadC-DNA complexes with a much lower mobility than freely diffusing CadC in the membrane.

Taken together, the existing data suggest that the one-component system CadC establishes the protein-DNA contact required for transcription regulation not by a conventional target search akin to cytoplasmic transcription factors (Fig 1A), but instead by 2D diffusion of the protein in the membrane and fluctuations of the DNA conformation that occasionally bring the DNA region of the target site close enough to the membrane to be captured by the protein (Fig 1B). Intuitively, a successful diffusion and capture event seems highly unlikely. However, that such events occur was independently demonstrated in an experiment that artificially tethered the Lac repressor to the cell membrane [28]. This construct was indeed able to inducibly repress transcription from a chromosomal reporter. Hence, the striking questions are how this type of target search is kinetically feasible and on which timescale. Here, we address these questions using *E. coli* CadC as a model system. We measure the search time of CadC to its target DNA-binding site in single cells by probing the formation of fluorescent CadC spots at different times. This yields experimental search time distributions that we compare against so-called 'first passage time' distributions obtained from stochastic models [29]. We also measure the mobility of a chromosomal locus in our experimental setup using a fluorescent repressor/

**A**  cytoplasmic transcription factor       **B**  membrane-integrated transcription factor

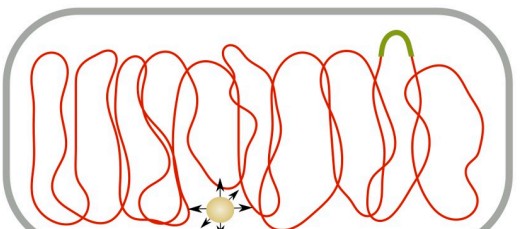 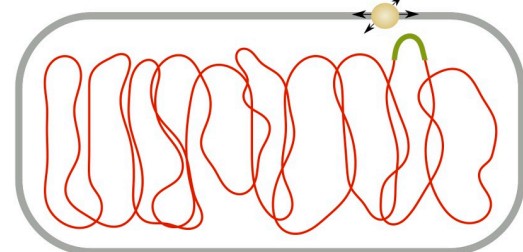

**Fig 1. DNA target search of cytoplasmic vs. membrane proteins. (A)** A cytoplasmic transcription factor (yellow) locates its specific binding site (green) on the DNA (red) through a combination of 3D diffusion in the cytoplasm and 1D sliding along the polymer. While the DNA itself is not static during this target search, DNA motion does not significantly contribute to its completion. **(B)** In contrast, a membrane-integrated transcription factor can only perform 2D diffusion in the membrane, such that DNA motion becomes essential. At a minimum, the specific DNA-binding site has to move close to the membrane to enable target recognition.

operator system to inform kinetic Monte Carlo simulations of the target search dynamics. These simulations are able to reproduce the experimental behavior and to elucidate properties of the search process that we cannot obtain experimentally.

## Results

### Choice of CadC as experimental model system

CadC is a particularly well studied membrane-integrated one-component receptor. It is part of a pH stress response system, which is also dependent on signaling input from the lysine-specific permease LysP [25]. The function of the Cad system is to alleviate acidic stress by activating synthesis of the lysine/cadaverine antiporter CadB and the lysine decarboxylase CadA, which converts lysine to cadaverine to be secreted by CadB. CadA and CadB are encoded by the *cadBA* operon, which is transcriptionally upregulated by specific binding of CadC upstream of the promoter [30], see Fig 2A (inset). CadC is known to respond to and integrate three different signals: It is activated by an acidic pH in the presence of external lysine and inhibited by cadaverine. The pH-sensory function as well as the feedback inhibition by cadaverine was assigned to distinct amino acids within the periplasmic sensory domain of CadC [26, 31], whereas the availability of external lysine is transduced to CadC via the co-sensor and inhibitor LysP, a lysine-specific transporter [32, 33]. High-affinity DNA-binding of CadC requires CadC homodimerization, which is inhibited by LysP via intramembrane and periplasmic contacts under non-inducing conditions [32, 33]. A drop in external pH induces dimerization of the periplasmic sensory domain of CadC followed by structural rearrangement of its cytoplasmic linker, permitting the DNA-binding domain of CadC to homodimerize [20, 31, 34]. The CadC protein number is extremely low (on average 1–3 molecules per cell [27]), mainly due to a low translation rate caused by polyproline stalling, which is only partially relieved by elongation factor P [35].

    In the present study, CadC serves as an experimental model system to investigate the DNA target search of a membrane-integrated transcription factor (Fig 2A). To probe the kinetics of this search process in individual cells, it is crucial to have a well-defined initial state and clear 'start' and 'stop' events. When cells are initially grown in a medium with neutral pH, CadC is inactive and homogeneously distributed in the membrane [27]. A sudden medium shift to low pH and rich lysine conditions then serves as a suitable 'start' trigger, which activates CadC via the signal transduction mechanism described above and starts the target search process.

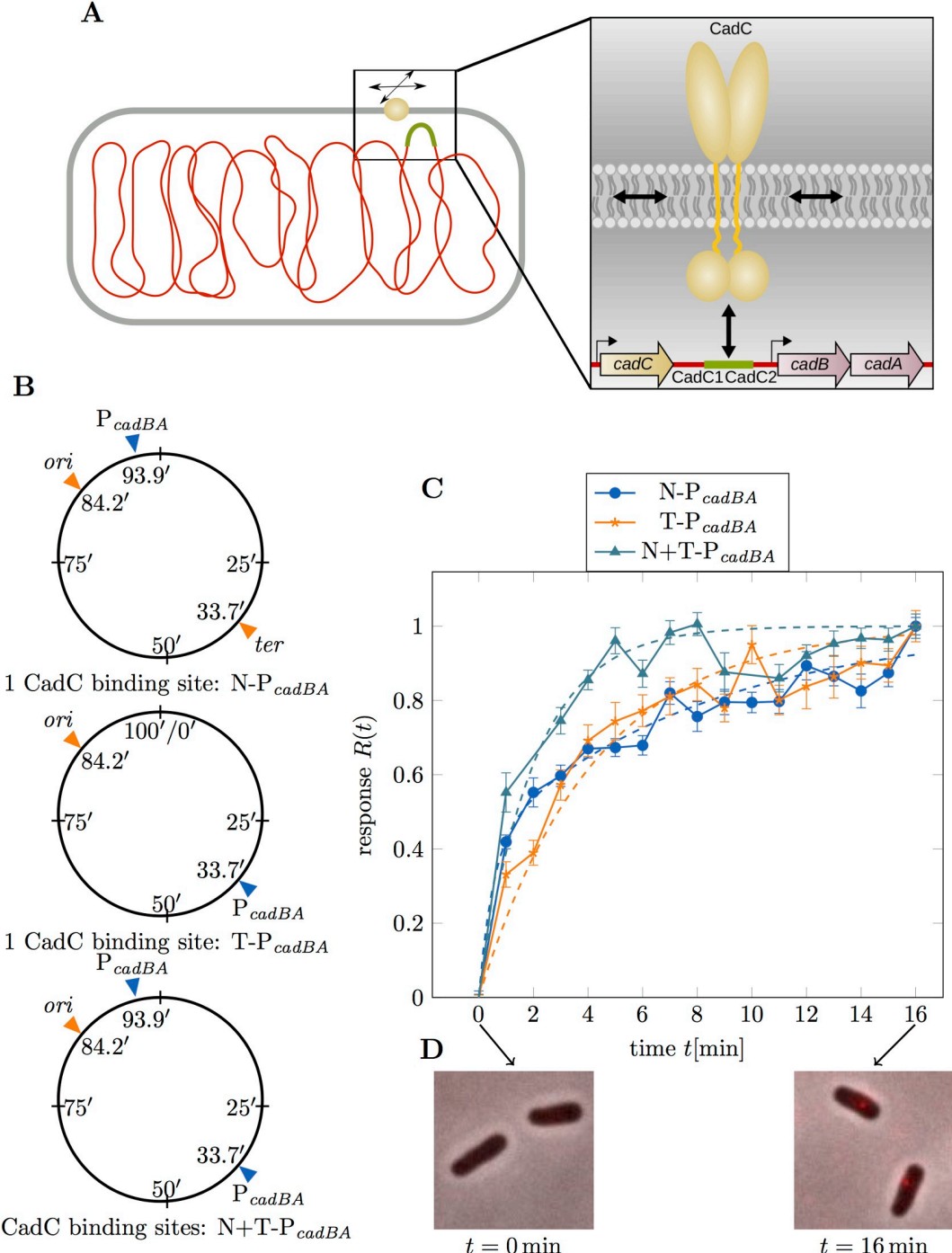

**Fig 2. Dynamics of the target search by CadC.** (**A**) The target search of membrane-integrated transcription factors is investigated by experimentally measuring the response time of CadC. The molecular model on the right shows CadC in yellow, diffusing in the membrane and forming two dimers to bind to the two CadC binding sites within the *cadBA* promoter (CadC1 and CadC2), displayed in green. The *cadC* gene is located upstream of the *cadBA* operon. (**B**) Three *E. coli* strains with different positions of the *cadBA* promoter: N-P*cadBA* with the native DNA-binding site close to *ori*, T-P*cadBA* with the binding site at the terminus and N+T-P*cadBA* with both binding sites. (**C**) Experimental results from CadC spot detection. Fluorescent microscopic images were taken every minute after receptor activation and analyzed for CadC spots for the strains defined in panel B. The plot shows the response $R(t)$, defined as the normalized fraction of cells with fluorescent spots as a function of time $t$ after exposure to acid stress. Error bars correspond to the propagated standard deviation of $v(t)$ from averaging over multiple data sets. The dashed lines in the plot show a fit of the response function to the CDF of a sequential reversible two-step model with mixed

initial condition for N-P$_{cadBA}$, shown in blue dots and with fixed initial condition for T-P$_{cadBA}$ (orange stars) and N+T-P$_{cadBA}$ (cyan triangles). (**D**) The fluorescence microscopy images demonstrate how fluorescent spots appear after receptor activation.

Dimerization of CadC is expected to occur on a much faster timescale than the protein-DNA search process (see 'Materials and methods'). Detecting the successful termination of the target search is challenging. Here, we exploit the previous finding that the formation of stable CadC-DNA complexes is visible as distinct spots in fluorescence microscopy images [27]. The study excluded that spot formation occurs solely due to low pH, using a pH-independent CadC variant that showed spots at both neutral and low pH. The connection between spot formation and CadC DNA-binding was derived from the observation that no spots were formed when CadC was rendered unable to bind DNA, and the number of spots per cell correlated with the number of DNA-binding sites for CadC. Without DNA-binding site only 20% of cells formed spots, possibly due to non-specific DNA-binding. We take this into account in our quantitative analysis below.

## Experimental measurement of CadC target search times

We used three *E. coli* strains with different binding site configurations on the chromosome: (i) N-P$_{cadBA}$ (wild type), with the native DNA-binding site at 93.9′, relatively close to the origin of replication (*ori*) [36], (ii) T-P$_{cadBA}$, with the binding site relocated to the terminus, and (iii) N +T-P$_{cadBA}$, with both binding sites (Fig 2B). To visualize the temporal and spatial localization of CadC *in vivo*, we transformed each of the strains with plasmid-encoded mCherry-tagged CadC, which slightly increases the average number of CadC molecules per cell to 3–5 [27]. After the medium shift at $t = 0$ min, we took fluorescence and phase contrast microscopy images of cells sampled from the same culture every minute. We used image analysis tools to detect fluorescent spots within the cells, evaluating between 859 and 2506 cells per time step, see 'Materials and methods'. Based on these data, we determined the fraction of cells with at least one fluorescent spot, $v(t) = N_{\text{cells with spot}}(t)/N_{\text{cells}}(t)$ at each time $t$ after CadC activation. To take into account the initial fraction of spots attributed to non-specific DNA-binding (see above), we defined the response function

$$R(t) = \frac{v(t) - v(0)}{v(\infty) - v(0)} \quad , \tag{1}$$

which rises from zero to one. Here, the asymptotic value $v(\infty)$ accounts for the fact that fluorescent spots are never detected in all cells (see the raw data in S1 Fig). This is likely due to the heterogeneous distribution of CadC [27]: Given the low average copy number, some cells are expected to have less than the two molecules required for dimerization. Additionally, some spots may have been missed by the spot detection algorithm, in particular for cells that were not perfectly in focus. The time-dependent response $R(t)$ for our three strains is shown in Fig 2C, with examples of fluorescence microscopy images of cells at $t = 0$ min and the last time point (Fig 2D). Before analyzing the experimental response functions, we discuss the description of the target search as a stochastic process and derive theoretical response functions for comparison with the experimental data.

## The target search as a stochastic process

To conceptualize the CadC target search process and the experimental response function, Eq 1, we turn to a coarse-grained model, in which the CadC search for a specific binding site on the DNA can be described by a stochastic process with a small number of discrete states. Since

both CadC and the DNA must move in order to establish a specific protein-DNA contact, it is reasonable to assume a reversible sequential process with an intermediate state,

$$S_1 \underset{k_2^-}{\overset{k_1^+}{\rightleftharpoons}} S_2 \overset{k_2^+}{\longrightarrow} S_3 \ . \tag{2}$$

Here, state $S_1$ corresponds to configurations where the DNA target site is not in direct vicinity of the membrane, while CadC is delocalized on the membrane, unbound to the DNA. State $S_3$ corresponds to the final state where CadC is bound to a specific target site on the DNA. The intermediate state $S_2$ could then correspond to configurations where the DNA segment containing the target site is close to the membrane, but CadC is not bound to this segment. The transition rates between these states are denoted as $k_1^+$, $k_2^+$, and $k_2^-$.

We are interested in the so-called 'first-passage time' $\tau$ to reach the final state $S_3$, which corresponds to the target search time within this coarse-grained description. The probability distribution $p(\tau)$ for this time is calculated by making the final state absorbing, using standard techniques [29] (see S1 Text). Assuming that the system is initially in state $S_1$, the first passage time distribution is

$$p(\tau) = \frac{e^{-\frac{\tau}{\alpha}} - e^{-\frac{\tau}{\beta}}}{\alpha - \beta} \ , \tag{3}$$

where the two timescales $\alpha$ and $\beta$ of the exponential functions are related to the transition rates via

$$\alpha \ = \ 2\left(k_1^+ + k_2^+ + k_2^- - \sqrt{\left(k_1^+ + k_2^+ + k_2^-\right)^2 - 4k_1^+ k_2^+}\right)^{-1} , \tag{4}$$

$$\beta \ = \ 2\left(k_1^+ + k_2^+ + k_2^- + \sqrt{\left(k_1^+ + k_2^+ + k_2^-\right)^2 - 4k_1^+ k_2^+}\right)^{-1} , \tag{5}$$

implying that $\alpha > \beta$. At large times, the distribution $p(\tau)$ decays exponentially (decay time $\alpha$), whereas the timescale $\beta$ corresponds to a delay at short times. Hence, increasing $\alpha$ leads to a slower decay and increasing $\beta$ to a longer delay.

To relate the first passage time distribution to our experimental response function, Eq 1, we consider the cumulative distribution function (CDF) defined by

$$\text{CDF}(\tau) = \int_0^\tau p(t)\mathrm{d}t \ , \tag{6}$$

which is the probability that the first passage time is less than or equal to $\tau$. Experimentally, $\text{CDF}(\tau)$ corresponds to the fraction of cells in which the target search was successful by time $\tau$. The response function in Eq 1 is our best proxy for this fraction of cells, and hence we identify

$$\text{CDF}(\tau) \ \hat{=} \ R(\tau) \ , \tag{7}$$

such that one can use the CDF of the reversible sequential process as a fitting function for our data (and similar experiments in the future). The cumulative distribution function for $p(\tau)$ of Eq 3 is

$$\text{CDF}(\tau) = 1 - \frac{\alpha e^{-\frac{\tau}{\alpha}} - \beta e^{-\frac{\tau}{\beta}}}{\alpha - \beta} \ . \tag{8}$$

However, using Eq 8 amounts to the assumption that all cells are initially in state $S_1$. If we

allow for the possibility that some cells are in state $S_2$ when CadC is activated, we have a mixed initial condition, where the process starts either from state $S_1$ or from state $S_2$. Denoting the fraction of cells that are initially in state $S_2$ by $x$, such that a fraction $1 - x$ starts in state $S_1$, we obtain a first passage time distribution and associated cumulative distribution function of the form

$$p(\tau) = \frac{e^{-\frac{\tau}{\alpha}} - e^{-\frac{\tau}{\beta}} + c\left(\alpha e^{-\frac{\tau}{\beta}} - \beta e^{-\frac{\tau}{\alpha}}\right)}{\alpha - \beta} \;,$$

$$\mathrm{CDF}(\tau) = 1 - \frac{\alpha e^{-\frac{\tau}{\alpha}} - \beta e^{-\frac{\tau}{\beta}} - \alpha\beta c\left(e^{-\frac{\tau}{\alpha}} - e^{-\frac{\tau}{\beta}}\right)}{\alpha - \beta} \;,$$

(9)

where the new parameter $c$ corresponds to $c = x k_2^+$.

We use Eqs 8 and 9 as fit functions to describe our experimental response functions. We obtain the average and the variance of the search times analytically from the fit parameters, including the statistical errors for both of these quantities (Table 1), using error propagation and the full statistical covariance matrix for the parameters, see S1 and S2 Tables and S1 Text. For instance, the average value of the distribution Eq 3, referred to as the mean first passage time, is

$$\langle \tau \rangle = \alpha + \beta = \left(1 + \frac{k_2^-}{k_2^+}\right)\frac{1}{k_1^+} + \frac{1}{k_2^+} \;.$$

(10)

This corresponds to the average time for the first step $(1/k_1^+)$ multiplied by the average number of trials needed to reach the second step, plus the average time for the second reaction step $(1/k_2^+)$. In cases where one of the reaction steps is rate limiting, the mean first passage time is simply the inverse of the limiting rate, and the process simplifies to a two-state process with an exponentially distributed first passage time. Similarly, for Eq 9, the mean first passage time is

$$\langle \tau \rangle = \alpha + \beta - \alpha\beta c \;.$$

(11)

## The mean search time is less than 5 min and not affected by relocation of the target site

We first analyzed the experimental response $R(t)$ in the wild type strain. The wild type data (N-P$_{cadBA}$) in Fig 2C (blue dots) show an initially fast increase, followed by a more gradual saturation. This behavior is not well described by Eq 8, but captured by Eq 9 with the mixed initial condition, as can be seen by the fit to the data represented by the blue dashed line in Fig 2C. From the fit parameters $\alpha$, $\beta$, and $c$, we computed the mean search time according to Eq

**Table 1. Means and variances of the search time distributions.**

| Strain | $\langle \tau \rangle$ [min] | $\sigma^2$ [min$^2$] |
|---|---|---|
| N-P$_{cadBA}$ | 4.84 ± 0.19 | 49.6 ± 8.8 |
| T-P$_{cadBA}$ | 4.20 ± 0.15 | 17.6 ± 2.0 |
| N+T-P$_{cadBA}$ | 2.02 ± 0.12 | 4.09 ± 0.66 |

Results from fitting the experimentally computed CDFs to the sequential reversible model with mixed initial condition (N-P$_{cadBA}$) and fixed initial condition (T-P$_{cadBA}$ and N+T-P$_{cadBA}$). The fit parameters $\alpha$, $\beta$ and $c$ were used to compute the mean first passage time $\langle \tau \rangle$ and the variance $\sigma^2$ with statistical errors obtained from error propagation using the full covariance matrix, see S1 and S2 Tables and S1 Text.

[11], finding $\langle\tau\rangle \approx 4.84 \pm 0.19$ min. This result is consistent with the transcriptional response of the target genes *cadBA*, which was previously probed by Northern blot analysis, finding that the cell-averaged *cadBA* mRNA level starts to increase about 5 min after receptor activation [25].

In order to assess whether the position of the DNA-binding site along the chromosome affects the target search, we analyzed the behavior of strain T-P*cadBA*, which has the CadC DNA-binding site at the terminus instead of the native position. The corresponding response function shown in Fig 2C (orange stars) features a less pronounced initial increase than observed for N-P*cadBA*, and the dashed orange line shows an adequate fit using the sequential model with fixed initial condition in Eq 8. The calculated mean first passage time of $\langle\tau\rangle \approx 4.20 \pm 0.15$ min is comparable to that for the wild type strain. The slower initial increase is compensated by a faster increase at later times to yield a slightly smaller mean search time. To characterize the shape of the mean first passage time distributions, we calculated the variance $\sigma^2$ of $p(\tau)$, see Table 1, which is smaller for strain T-P*cadBA* than for the wild type.

## Search time is decreased with two chromosomal CadC binding sites

After observing essentially the same mean search time for two very distant locations of CadC target sites on the chromosome, we wondered how a strain harboring both target sites would behave. We therefore repeated the measurements for *E. coli* strain N+T-P*cadBA*, which has the native DNA-binding site and additionally the binding site at the terminus. As shown in Fig 2C (cyan triangles), the response function of this strain saturates much earlier than for the other two strains. Fitting the response data to the sequential model with fixed initial condition (Eq 8), we obtained a mean first passage time of $\langle\tau\rangle \approx 2.02 \pm 0.12$ min, which is only around half of the time than for a single chromosomal binding site.

## Colocalization of CadC spots with the DNA-binding site

We also wondered whether the fluorescence spots indicating the position of stable CadC-DNA complexes in single cells would show a similar spatial distribution as the *cadBA* locus. We therefore analyzed the localization of CadC spots along the long axis of the cell in *E. coli* wild type. As an estimate for the position of the *cadBA* locus along the cell, we tracked the position of *ori* at low pH. Towards this end we inserted a *parS* gene close to *ori* and let ParB-yGFP bind to it, making the *ori* region visible as a fluorescent spot in microscopy images. As the position of chromosomal loci depends on the progression of the cell cycle [37], we grouped cells according to their length into three classes. Fig 3 shows the spatial distribution of the relative spot positions along the half long axis for these three length classes, comparing ParB spots (blue) to CadC spots (orange). The large overlap of the distributions implies a similar cell age dependent localization of *ori* and CadC spots along the long cell axis, suggesting that CadC spots indeed form close to the DNA-binding site.

## Biophysical model of the target search process

To gain more insight into the dynamics of the target search, we turned to a coarse-grained biophysical model for the coupled dynamics of CadC and the DNA. We simulated the search of a CadC dimer for its target DNA-binding site using a lattice model and a kinetic Monte Carlo approach. As depicted in Fig 4A, the DNA is represented by a path on the 3D lattice (the 'cytoplasm') and the CadC dimer by a point on the surface of the lattice (the 'membrane'). Starting from a random initial configuration, the DNA moves in the cytoplasm and the CadC dimer diffuses in the membrane until it reaches the specific DNA-binding site to end the search process, see 'Materials and methods'. As illustrated in Fig 4B, CadC dimer diffusion is

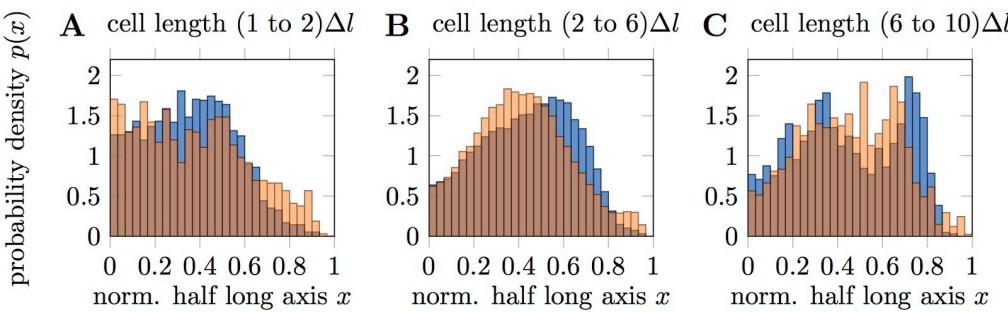

**Fig 3. Localization of *ori* and CadC spots.** Localization probability of CadC spots in N-P*cadBA* cells (orange) and ParB localization marking *ori* (blue) along the half long axis of cells. The half long axis is normalized such that mid-cell is at $x = 0$ and the poles are at $x = 1$. Overlaps of the two distributions are shown in darker orange. Cell age is taken into account by splitting all occurring cell lengths into ten equally spaced steps $\Delta l$ and pooling the cells according to their size. From the ten different age classes we observed similar localization probabilities for $l = (1 \text{ to } 2)\Delta l$ (**A**), $l = (2 \text{ to } 6)\Delta l$ (**B**) and $l = (6 \text{ to } 10)\Delta l$ (**C**), which are therefore grouped together in this plot.

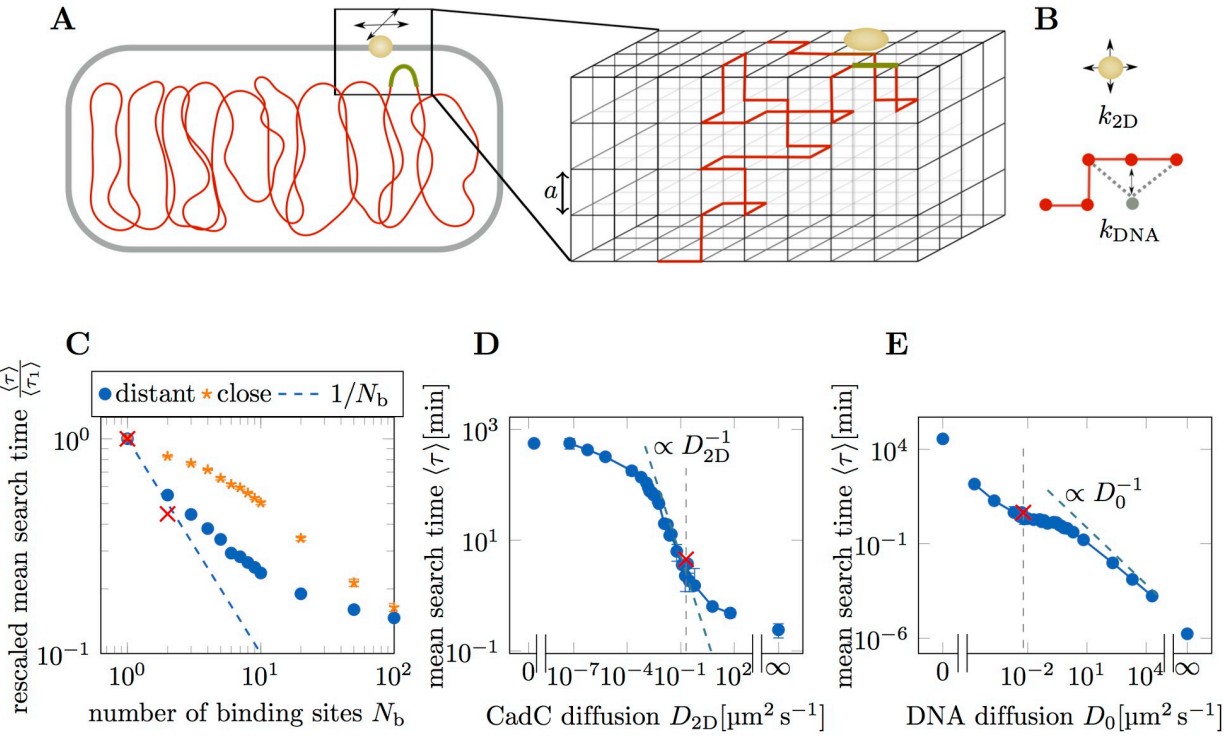

**Fig 4. Characterization of the target search by computer simulation.** (**A**) The target search of membrane-integrated transcription factors is studied by kinetic Monte Carlo simulations of a lattice model. The movement of a CadC dimer (yellow) is restricted to the surface of the lattice, whereas the segments of the DNA contour (red) can be anywhere within the lattice. (**B**) CadC dimers hop to neighboring surface lattice sites with a hopping rate $k_{2D}$, while the DNA contour moves by random displacements of single vertices ("beads") with a basal rate $k_{DNA}$, see 'Materials and methods'. (**C**) Dependence of the mean search time $\langle \tau \rangle$, normalized by the search time with a single binding site $\langle \tau_1 \rangle$, on the number of DNA-binding sites $N_b$. Realistic parameters are used for binding sites placed either uniformly (blue dots) or next to each other (orange stars) along the circular chromosome. Red crosses correspond to the experimental results for one (N-P*cadBA*) and two (N+T-P*cadBA*) DNA-binding sites. (**D**) Mean target search time as a function of the CadC diffusion coefficient. The leftmost data point corresponds to $k_{2D} = 0$ and the rightmost data point corresponds to $k_{2D} \to \infty$, simulated by making the whole cell surface the target. The estimated diffusion coefficient of a CadC dimer, $D_{2D} = 0.20 \ \mu m^2 \ s^{-1}$ is marked (gray dashed line), as well as the experimentally measured search time for N-P*cadBA* (red cross). (**E**) Mean target search time as a function of the DNA diffusion coefficient. The dashed gray line marks $D_0 = 0.0060 \ \mu m^2 \ s^{-1}$, used in the realistic parameter set, and the red cross marks the experimentally measured search time for N-P*cadBA*.

**Table 2. Simulation parameters matching the experiments.**

| Parameter | Symbol | Value |
|---|---|---|
| lattice constant | $a$ | 100 nm |
| cell volume | $V_{\text{cell}}$ | 1 µm$^3$ |
| DNA length [beads] | $N_{\text{DNA}}$ | 100 |
| number of CadC dimers | $N_{\text{p}}$ | 1 |
| number of binding sites | $N_{\text{b}}$ | 1 |
| CadC diffusion coeff. | $D_{\text{2D}}$ | 0.20 µm$^2$ s$^{-1}$ |
| DNA diffusion coeff. | $D_0$ | 0.0060 µm$^2$ s$^{-1}$ |

Parameter set for the kinetic Monte Carlo simulations that were chosen to match the experiments.

implemented by random hopping to neighboring surface lattice sites with a rate $k_{\text{2D}}$, and the DNA dynamics is implemented by random displacements of single beads at a basal rate $k_{\text{DNA}}$, see 'Materials and methods'. For each set of parameters we run $\geq 10^3$ simulations to compute mean values and search time distributions.

To compare our target search simulations to our experiments we used parameter values based on experimental estimates, as discussed below and in 'Materials and methods', and summarized in Table 2. In the following, we refer to these parameter values as the 'realistic parameter set'. Since the move rates for CadC dimers and the DNA are based on *in vivo* measurements, they implicitly take into account effects due to crowding in the cytoplasm and in the membrane. Instead of simulating the full length of an *E. coli* chromosome, we made use of an observed relation between search time and polymer length to reduce computation time. After an initial increase of the mean search time with the number of DNA segments, it becomes independent of polymer length as it reaches a plateau (see S3 Fig). Hence it is not necessary to simulate the full length of an *E. coli* chromosome, as long as the simulated polymer length is in the plateau region. For the realistic parameter set used in S3 Fig, a polymer length of $N_{\text{DNA}} = 100$ beads is sufficient, which is the minimum value we used in our simulations.

## Mobility measurements of the DNA

To complete our parameter set, we estimated the diffusion constant $D_0$ of a DNA segment in our simulations by experimentally tracking a chromosomal locus. The origin was tagged using the same *parS*/ParB fluorescent operator/repressor system (FROS) as discussed above. For N-P$_{cadBA}$-*parS_ori* and Δ*cadC-parS_ori*, a wild type and a mutant *E. coli* strain lacking *cadC*, each containing *parS_ori*, fluorescence and phase contrast microscopy time lapse videos were taken of the same cells every 30 s for activating and inactivating conditions, respectively.

The mean square displacement (MSD) was obtained by selecting the closest spots in subsequent image frames and calculating the ensemble-averaged MSD. The resulting MSD curves (S2 Fig) are in semi-quantitative agreement with other tracking experiments of chromosomal loci in *E. coli* [38, 39]. Previous experiments showed that DNA diffusion in *E. coli* agrees well with the Rouse model [40], with diffusion exponents in the range 0.4–0.6 [38, 39, 41]. We therefore fitted the mean square displacement curves to MSD$(\tau) = \Gamma\tau^{0.5}$, obtaining $\Gamma = 0.0111 \pm 0.0001$ µm$^2$ s$^{-0.5}$ for *E. coli* wild type under activating conditions and $\Gamma = 0.0091 \pm 0.0001$ µm$^2$ s$^{-0.5}$ for Δ*cadC* under inactivating conditions. The DNA mobility seems to be independent of the probed conditions, since the two values do not differ significantly. Hence we used the average value $\langle\Gamma\rangle$ to compute the diffusion constant $D_0$ of a Kuhn segment according to the Rouse model, see 'Materials and methods'.

## Simulating multiple DNA-binding sites

Motivated by the experiments with the N+T-P$_{cadBA}$ strain, we performed simulations with varying number of DNA-binding sites. In Fig 4C the mean search time $\langle \tau \rangle$ normalized by the mean search time with a single DNA-binding site $\langle \tau_1 \rangle$ is shown as a function of the number of binding sites $N_b$. Placing the binding sites as far apart as possible on the circular chromosome, simulations with realistic parameters show a halving in search time when increasing the number of binding sites from one to two. This is the expected result for two binding sites moving independently of each other due to the decorrelation of polymer subchains in spatial confinement. How far two binding sites have to be apart along the DNA to behave as independent targets therefore depends on the size of the simulated cell. The initial inverse-$N_b$ scaling of $\langle \tau \rangle$ flattens progressively as the binding sites come closer to one another and are more correlated in their movement. Placing the binding sites next to each other (orange stars) has an expectedly small effect for small $N_b$, since it only increases the size of the binding site. The curve becomes steeper as the binding sites occupy a larger fraction of the polymer.

Given that the two DNA-binding sites in N+T-P$_{cadBA}$ are on opposite sides of the chromosome, we expect them to behave as two independent binding sites. The experimentally measured reduction of the search time by roughly a factor of two (red crosses in Fig 4C) is therefore in good agreement with our model. Also, by construction, the position of the binding site along the chromosome has no effect on the simulated search times.

## Target search time is sensitive to CadC diffusion

To address the question whether the search process is more sensitive to changes in DNA mobility or CadC diffusion, we performed simulations with varying diffusion rates. Plotting the mean search time as a function of the CadC diffusion coefficient, $D_{2D}$, in Fig 4D, we observe three different regimes. For slow protein diffusion, the search time is almost independent of $D_{2D}$, followed by a range where the search process is entirely dominated by CadC diffusion ($\tau \propto D_{2D}^{-1}$), while for very fast CadC diffusion the search becomes less sensitive to $D_{2D}$ again. The data point for infinitely fast CadC diffusion was obtained by making CadC cover the whole cell surface. The realistic value for the diffusion constant of CadC ($D_{2D} = 0.2 \ \mu m^2$ $s^{-1}$) is marked by a gray dashed line and lies in the regime where the search time strongly depends on $D_{2D}$. The red cross marks the experimentally measured mean search time for a single binding site, which is in surprisingly good agreement with the simulation data, given the simplicity of our biophysical model. Fig 4E shows the counterpart of Fig 4D for DNA diffusion, with $D_0 = 0.006 \ \mu m^2 \ s^{-1}$ marked by a gray dashed line. While the mean search time is strongly dependent on DNA diffusion for small and large $D_0$, it is less sensitive to $D_0$ in the experimentally relevant intermediate regime.

To further analyze this observation, we used our simulations to approximate a target search exclusively due to CadC diffusion in the membrane. We used the realistic parameters but placed the DNA-binding site at the membrane and set the DNA diffusion rate to zero, such that only CadC was moving, yielding a mean search time of $\langle \tau \rangle \approx 18$ s. This is consistent with an estimate based on the previously reported [42] approximate formula, $\tau \approx \frac{R_s^2}{D} \left( -2 \ln \left[ \frac{r_t}{R_s} \right] + 2 \ln [2] - 1 \right)$, for the mean first encounter time of a particle moving with diffusion constant $D$ on the surface of a sphere with radius $R_s$ with a trap of radius $r_t$. Approximating the cubic cell by a sphere of radius $R_s = 0.69 \ \mu m$ and CadC with $r_t = 0.050 \ \mu m$, this yields $\tau \approx 13$ s. This estimate also validates our simulations by showing that the dependence of the search time on the size of CadC is weak, rendering a correction for overestimating the size of CadC dimers in the simulations unnecessary.

While the DNA is likely to be the less mobile part in the target search, it has to move at least close to the membrane to enable binding to CadC. The data point on the very right of Fig 4D corresponds to the time it takes the DNA-binding site to bind anywhere to the membrane. It corresponds to $\tau \approx 14$ s, similar to the time it takes CadC to locate the binding site at the membrane. Therefore, a scenario where the DNA-binding site randomly reaches the membrane and CadC searches the membrane to bind to it seems to lead to reasonable search times according to our simulations.

For comparison we also simulated a target search process where CadC is immobile and DNA diffusion has to account for the whole search. When only the DNA is moving and all other rates are set to zero, with realistic values for DNA length and cell size a mean search time of $\langle \tau \rangle \approx 560$ min was calculated from the simulations, a response time that would not allow *E. coli* cells to survive the transition to acidic environments. Using the approximate formula $\langle \tau \rangle \approx \frac{\Omega}{4 r_t D}$ [43] for the encounter time of a particle with diffusion constant $D$ to locate a target of radius $r_t$ on the surface of a confining spherical domain with volume $V$ we obtained $\tau \approx 14$ min for a monomer with diffusion constant $D_0$ and $\tau \approx 1400$ min for a monomer with diffusion constant $\frac{D_0}{N_{DNA}}$. Finding a value between these two approximations is what we expected, as they do not account for the polymer dynamics.

## Quantitative comparison of simulated and experimental search time distributions

As shown above, the numerical simulations of the target search using our experimentally estimated parameter set yield mean search times that are fairly close to what we measured experimentally. To test whether the biophysical model can also quantitatively capture the experimental behavior, we attempted to find simulations with time distributions closely matching the experimentally computed CDFs. In Fig 5A the experimental CDF of N-P$_{cadBA}$ together with the best fit to the sequential model with mixed initial condition is shown in orange. In order to find a simulation with agreeing time distributions, we simulated the target search process using the estimated parameter set but improved the agreement by increasing the simulated cell volume to $V_{cell} = 1.3$ μm$^3$. As shown by the blue dots in Fig 5A this yields a very good accordance between experimental data and simulations. In Fig 5B the experimental CDF of N+T-P$_{cadBA}$ and the corresponding fit to the sequential model with fixed initial condition is shown in orange. Our attempt to find a matching simulation by using the same parameters as for the simulations in panel A but increasing the DNA-binding sites to $N_b = 2$ leads to quite a good agreement with the experiment, as shown by the blue dots. Despite the simplicity of the simulations, neglecting in particular constraints of DNA movement due to its specific organization in the cell, the results agree surprisingly well with the experimental findings when using experimentally realistic parameters.

To further investigate the simulated first passage time distributions, we computed them for different parameter settings and fitted to the sequential model with fixed initial condition (Eq 8) and with mixed initial condition (Eq 9) respectively. While the shape of the first passage time distribution is independent of the system size parameters, the dynamic parameters have a big effect. For most parameter settings, including the realistic parameters, we found the best agreement of simulated CDFs with the sequential model with mixed initial condition. However, for simulations with an immobile polymer or slowly diffusing CadC, the search time distribution agrees rather with the sequential model with fixed initial condition. Since both in the experimental and simulated CDFs the delay $\beta$ is very small compared to $\alpha$, a fit to the fixed initial model gives virtually the same result as a single exponential CDF. When either the DNA or

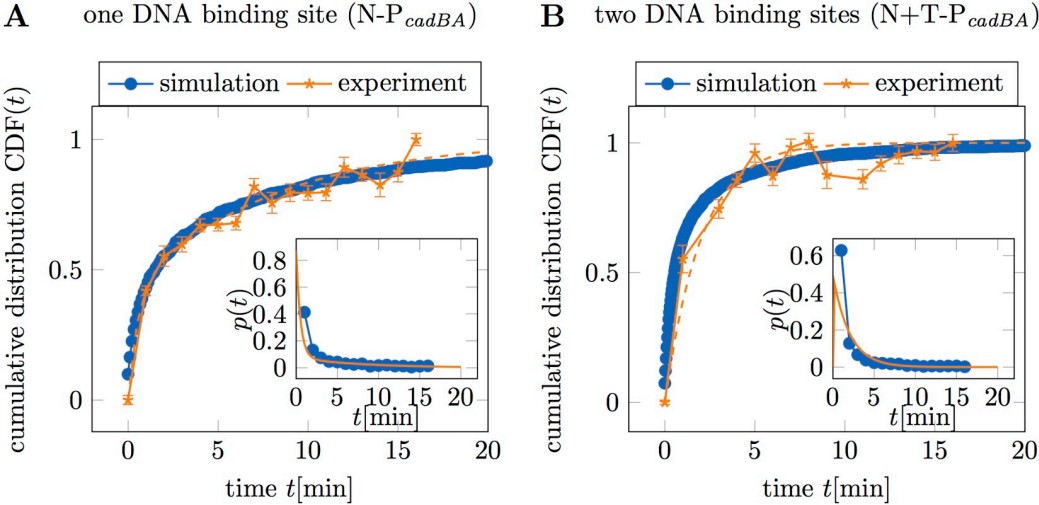

**Fig 5. Matching simulations to the experimental results.** Numerical simulations are directly compared to the experimental distributions. Experimental data are shown in orange stars, the orange dashed line corresponds to the best fit. Blue dots correspond to the simulations that agree most with the experimental data. Insets depict the corresponding first passage time distributions. (**A**) The experimental CDF of N-P$_{cadBA}$ is shown together with the best fit and simulations using the realistic parameter set, summarized in Table 2. To yield better agreement with the experimental CDF the cell volume was increased to $V_{cell} = 1.3 \ \mu m^3$. (**B**) The experimental CDF of N+T-P$_{cadBA}$ is shown together with the best fit and simulations using the parameter set matching the experimental values, with cell volume $V_{cell} = 1.3 \ \mu m^3$ and two binding sites on the polymer. All simulations were started from a random initial configuration.

CadC accounts for most of the search there is only one limiting rate in the process, which can therefore be described by a single exponential distribution.

## Discussion

We combined fluorescence microscopy experiments, quantitative analysis, and kinetic Monte Carlo simulations to characterize the target search kinetics of membrane-integrated transcription factors for a specific binding site on the chromosomal DNA. We were able to measure the time between the environmental stimulus and stable DNA-binding of a membrane-integrated one-component receptor, using the pH stress-sensing receptor CadC in *E. coli* as a model system. The measured mean search time of on average 4.5 min for a single DNA-binding site is consistent with the timescale of the earliest transcriptional response [25]. Given the severe constraint of a membrane-anchored target search, it seems surprising that the search time is only about 5-fold slower than the search time of the cytosolic Lac repressor for its operator, which takes around one minute at a similarly low protein level [12].

As the position of the DNA-binding site along the chromosome has no influence on the mean search time, the target search process appears to be quite robust. Given that the chromosome is highly organized within the cell, we believe this effect to arise mostly due to the mobility of CadC. While we do not have new evidence against proteolytic processing, this leads us to favor a diffusion and capture mechanism over transertion. Diffusion and capture mechanisms are well established for the localization of membrane-integrated proteins like SpoIVB in *Bacillus subtilis* [44, 45]. As the search time is independent of the position of the DNA-binding site, transertion of CadC is at least not a requirement for fast response, in agreement with a previous evaluation of the three models [27]. Our finding that the mean search time decreases by a factor of two in a mutant with two DNA-binding sites is also consistent with the diffusion and capture mechanism. Our simulations show that this is the expected result for two independent

and equally accessible binding sites, where distant parts of the polymer become uncorrelated in their motion due to the confinement in the cell.

Despite the simplicity of our biophysical model of the target search, we obtained search times that match the experimental measurements surprisingly well. Our simulations with a fixed protein number (one CadC dimer) produced search time distributions that were compatible with the experimental ones both in terms of their mean values and their width, suggesting that cell-to-cell heterogeneity in the number of CadC dimers does not significantly broaden the search time distribution. The constraint of chromosomal loci to macrodomains, which is neglected by our model, likely does not play a major role on the relatively short timescale of the search process, given that lateral motion of the CadC locus into the vicinity of the membrane is sufficient. Although we did not explicitly model crowding effects in the cytoplasm and at the membrane, the general slowdown of the kinetics due to crowding should be accounted for by our *in vivo* diffusion coefficients. Another open question is the lifetime of CadC dimers in the membrane. Active degradation should be negligible compared to the timescale of the search process. The majority of proteins in *E. coli* are not subject to rapid degradation [46] and individual membrane proteins that have been studied in detail displayed life-times as long as 6 days [47]. Spontaneous separation of CadC dimers from the membrane (or proteolytic cleavage) also do not appear to play a role, since CadC is deactivated by external cadaverine [25].

While it is difficult to experimentally answer the question whether the timescale of DNA motion or CadC diffusion limits the search process, the consistency between our simulations and experimental findings allows to draw some conclusions from an analysis of the simulations. In the experimentally relevant regime, the simulated mean search time is inversely proportional to the diffusion coefficient of the CadC dimer, but is relatively insensitive to the DNA diffusion speed. The search process therefore seems to be predominantly limited by the mobility of the transcription factors. Since within the typical range of diffusion coefficients of membrane-integrated proteins [48, 49], the mean search time varies inversely proportional with the diffusion coefficient, this result is expected to hold for DNA-binding membrane proteins in general.

Beyond mean values the experimental data allowed the extraction of the cumulative distribution functions (CDF) for the target search time. Describing them by the distribution obtained from a reversible sequential two-step process resulted in a good fit for the wild type data assuming a mixed initial condition. The same was true for our simulations of the target search process with most parameter settings. Using a parameter set estimated for the experimental conditions led to a distribution that matched the experimental CDF of *E. coli* wild type. Attempting to find a matching simulation for the mutant with two DNA-binding sites by using the same parameters with two DNA-binding sites also yielded a good agreement. In contrast to the simulations, however, the experimental CDF of the mutant with two binding sites and the mutant with the binding site at *ter* show a better agreement with the sequential model with fixed initial condition (or a single-exponential CDF). How exactly the position of the DNA-binding site affects the initial condition of the search process is not clear at this point. In the simulations, distributions agreeing with the sequential model with fixed initial conditions are obtained when either only CadC is moving, or it is moving very slowly.

Taken together, our experiments and simulations indicate that CadC is highly mobile in the membrane, while the *cadBA* promoter on the *E. coli* chromosome is mobile enough to randomly reach the membrane, enabling CadC to locate the DNA-binding site within about five minutes, independent of its position along the chromosome. While diffusion and capture mechanisms are established for the polar localization of membrane proteins [44, 45, 50], our study indicates a broader relevance of diffusion and capture mechanisms for the largely uncharacterized interactions of membrane-integrated proteins with chromosomal DNA [22].

## Materials and methods

### Model and simulations

Since the target search of CadC for its DNA-binding site(s) has been experimentally shown to succeed within a few minutes, we were aiming at finding the simplest model that can rationalize this fast response. We constrained our simulations to the main components of the target search: The DNA with one or more specific binding sites moving inside an *E. coli* cell and one or more CadC beads diffusing in the cell membrane. CadC dimerization was not simulated explicitly, instead the CadC beads on the cell surface correspond to already formed dimers. To justify this, we estimate the time of dimerization for two CadC monomers using the approximate formula [42]

$$\tau \approx \frac{R_s^2}{D} \left( -2 \ln \left[ \frac{r_t}{R_s} \right] + 2 \ln [2] - 1 \right) ,$$

(12)

for the mean time for a particle moving with diffusion constant $D$ on the surface of a sphere of radius $R_s$ to first encounter a trap of radius $r_t$. To estimate the time of dimerization for two moving CadC monomers with diffusion constant $D_1$ and radius $r_1$, we insert the relative diffusion constant $D = 2D_1$ and the contact radius $r_t = 2r_1$. Using a sphere of the same typical surface area as a cell, 4.40 $\mu m^2$ [51], we obtain $R_s = 0.59$ $\mu m$. For the radius of a CadC monomer we use $r_1 = 2.40$ nm [52] and for the CadC monomer diffusion constant $D_1 = 0.34$ $\mu m^2$ $s^{-1}$, which is a typical value for a membrane protein with a single transmembrane domain [53]. We then obtain a mean first encounter time of 5.1s, much faster than the total observed response time of several minutes and therefore allowing us to omit this step in our biophysical model of the target search.

To simulate a system as large as the whole genome in an *E. coli* cell for a time span long enough to capture the target search of CadC for its DNA-binding site is computationally challenging. We therefore chose a coarse-grained cubic lattice model in order to reduce the simulation time. Since we are interested in the search times, i.e. long timescales that involve many steps on the grid, the discrete description is a good approximation. As we take the average over different initial conditions, the small fraction of initial states that are already close to the target state and for which the search time is overestimated by the approximation should be negligible given our typical lattice sizes. In the simple cubic lattice model, the cell is composed of $N_{cell}^3$ lattice points forming a cuboid. The lattice constant was chosen equal to the Kuhn length $l_K = 2l_p$ of double stranded DNA (persistence length $l_p \approx 50$ nm [54]), such that the polymer could be represented as a freely jointed chain (FJC) on the lattice. It is composed of $N_{DNA}$ beads with coordinates $r = (r_1, \ldots, r_{N_{DNA}})$ placed on the grid points connected by $N_{DNA} - 1$ bonds. One or more of the beads is specified as the specific binding site(s). A CadC dimer occupies a single lattice point on the cell surface. In analogy to the single-site Bond-Fluctuation model [55], the polymer beads are connected by bonds of variable length, and are free to occupy also diagonal configurations. As we simulate a phantom chain, we replace the upper and lower bound of the bond length intended to ensure excluded volume and prevent chain crossings [55] by a spring potential

$$U\{r_n(t)\} = \sum_{n=1}^{N_{DNA}-1} k_s(|r_{n+1}(t) - r_n(t)| - b)^2 ,$$

(13)

with spring constant $k_s$ and an equilibrium bond length $b = l_K$. As in the Bond-Fluctuation model, the polymer moves by displacements of single beads, which can hop to one of the nearest lattice points. A CadC dimer moves by random hopping to neareast-neighbor lattice points

on the surface of the lattice. These kinetics are governed by the master equation

$$\frac{dp(\boldsymbol{x}, t)}{dt} = \sum_{\boldsymbol{x'}} [w_{\boldsymbol{x'x}} p(\boldsymbol{x'}, t) - w_{\boldsymbol{xx'}} p(\boldsymbol{x}, t)] \ , \tag{14}$$

with $p(\boldsymbol{x}, t)$ the probability that the system is in state $\boldsymbol{x}$ at time $t$. A state $\boldsymbol{x}$ defines both the conformation of the polymer and the position of the CadC dimer. The transition rates $w_{\boldsymbol{x'x}}$, to go from state $\boldsymbol{x'}$ to state $\boldsymbol{x}$ are only nonzero when the two states differ by a single move, i.e. the position of a single DNA bead or the position of the CadC dimer. The transition rate for moving CadC is obtained from the diffusion coefficient of CadC, $D_{2D}$, and the lattice constant $a$ as $k_{2D} = D_{2D}/a^2$. The rate of moving a DNA bead depends on the spring potential as in the Metropolis algorithm [56], i.e. as $k_{DNA} \cdot \min(1, \exp[-U/k_B T])$, where $U$ is the configuration-dependent potential energy according to Eq 13. The rate constant $k_{DNA}$ is adjusted numerically such that the free polymer (on an infinite lattice) displays a center of mass diffusion coefficient of $D_0/N_{DNA}$, where $D_0$ is the diffusion coefficient extracted from our DNA tracking data as described below. For ease of implementation we simulated the master equation with the null-event algorithm described in [57]. Unless otherwise stated the simulations were initialized with a random walk configuration of the polymer inside the cell and a randomly placed CadC bead in the membrane. To compute mean values and search time distributions $\geq 10^3$ simulations were run for every choice of parameters, each starting from a different initial configuration.

We estimated our 'realistic parameter set' as follows (see Table 2 for a summary of the values). The *E. coli* cell volume of $\propto 1 \ \mu m^3$ is approximated by a simulation box of volume $N_{cell}^3$ with $N_{cell} = 10a$ and lattice constant $a = 100$ nm. The *E. coli* chromosome with 4 639 221 bp [58], which measures 1.58 mm corresponds to a length of 15773$a$. Since our simulations have shown that the target search dynamics are independent of polymer length once it has reached a cell size dependent critical length (see S3 Fig), we save computation time by choosing the polymer length well above this threshold at 100$a$. It has been reported that the diffusion coefficient for protein diffusion within the cell membrane depends primarily on the size of the transmembrane domain [48, 49]. A CadC dimer has two transmembrane helices like the membrane protein WALP-KcsA, for which the diffusion constant was measured in multiple studies, yielding values between 0.21 $\mu m^2 \ s^{-1}$ [49] and 0.25 $\mu m^2 \ s^{-1}$ [59]. In a recent study the relevant CadC diffusion coefficient was also measured, albeit with a different fluorophor (mNG) than in the present study [27]. Those measurements yielded values ranging from 0.07 $\mu m^2 \ s^{-1}$ to 0.19 $\mu m^2 \ s^{-1}$. For our estimate of the model parameter values, we used the rounded value of $D_{2D} = 0.2 \ \mu m^2 \ s^{-1}$, which lies in the range of the quoted values. From our *ori* tracking experiments we calculated the diffusion constant of a DNA bead in our simulation to be $D_0 \approx 0.0060 \ \mu m^2 \ s^{-1}$, which is the value we chose for the realistic parameter set. To convert the dimensionless search times $\tau'$ from the simulations into seconds we computed $\tau' \frac{a^2}{D_0} \approx 1.66 \tau'$.

## Experiments

**Construction of strains and plasmids.** Molecular methods were carried out according to standard protocols or according to the manufacturer's instructions. Kits for the isolation of plasmids and the purification of PCR products were purchased from Süd-Laborbedarf (SLG; Gauting, Germany). Enzymes were purchased from New England BioLabs (Frankfurt, Germany). Bacterial strains and plasmids used in this study are summarized in Table 3.

*E. coli* strains were cultivated in LB medium (10 g l$^{-1}$ NaCl, 10 g l$^{-1}$ tryptone, 5 g l$^{-1}$ yeast extract) or in Kim Epstein (KE) medium [60] adjusted to pH 5.8 or pH 7.6, using the

**Table 3. Strains and plasmids used in this study.**

| Strains | Relevant genotype or description | Reference |
|---|---|---|
| *E. coli* DH5αλpir | *endA1 hsdR17 glnV44 (= supE44) thi-1 recA1 gyrA96 relA1 Ψ80'lacΔ (lacZ)M15 Δ((lacZYA-argF)U169 zdg-232::Tn10 uidA::pir*⁺ | [62] |
| *E. coli* WM3064 | *thrB1004 pro thi rpsL hsdS lacZΔM15 RP4-1360 Δ(araBAD)567 ΔdapA1341::[erm pir]* | [63] |
| *E. coli* MG1655 (N-P$_{cadBA}$) | K-12 F⁻ λ⁻ *ilvG⁻ rfb-50 rph-1* | [58] |
| *E. coli* MG1655_ P$_{cadBA}$_terminus (N+T-P$_{cadBA}$) | Additional *cadBA* promoter region at the terminus (33.7′) in MG1655 | This work |
| *E. coli* MG1655ΔP$_{cadBA}$_ P$_{cadBA}$_terminus (T-P$_{cadBA}$) | Clean deletion of *cadBA* promoter region in MG1655 with relocated *cadBA* promoter region at the terminus (33.7′) | [27] |
| *E. coli* MG1655-*parS_ori* | Insertion of the *Yersinia pestis* pMT1*parS* site at the *ori* (84.3′) in MG1655 | This work |
| *E. coli* MG1655Δ*cadC* | Deletion of *cadC* gene in MG1655, Km$^R$ | [64] |
| *E. cadC-parS_ori* | Insertion of the *Yersinia pestis* pMT1*parS* site at the *ori* (84.3′) in MG1655Δ*cadC*, Km$^R$ | This work |
| **Plasmids** | **Relevant genotype or description** | **Reference** |
| pET-*mCherry-cadC* | N-terminal fusion of *cadC* with *mCherry*, connected with a 22 amino acid long linker containing a 10His tag in pET16b, Amp$^R$ | [27] |
| pNTPS138-R6KT-P$_{cadBA}$_terminus | pNPTS-138-R6KT-derived suicide plasmid for insertion of *cadBA* promoter region at terminus in *E. coli* MG1655, Km$^R$ | [27] |
| pNTPS138-R6KT-*parS_ori* | pNPTS-138-R6KT-derived suicide plasmid for insertion of the *Yersinia pestis* pMT1*parS* site at the *ori* in *E. coli* MG1655, Km$^R$ | This work |
| pFHC2973 | N-terminal fusion of *parB* with *ygfp*, Amp$^R$ | [65] |
| pFH3228 | Plasmid carrying the pMT1-*parS* of *Yersinia pestis*, Amp$^R$ | [65] |

corresponding phosphate-buffer. *E. coli* strains were always incubated aerobically in a rotary shaker at 37°C. KE medium was always supplemented with 0.20% (w/v) glucose. Generally, lysine was added to a final concentration of 10 mmol unless otherwise stated. If necessary, media were supplemented with 100 μg ml⁻¹ ampicillin or 50 μg ml⁻¹ kanamycin sulfate. To allow the growth of the conjugation strain *E. coli* WM3064, we added meso-diamino-pimelic acid (DAP) to a final concentration of 200 μmol.

In order to gain strain *E. coli* MG1655-*parS_ori*, the *parS* site of *Yersinia pestis* was inserted close to the *ori*, at 84.3′ in *E. coli* MG1655. Briefly, the *parS* region was inserted between *pstS* and *glmS*. Therefore, DNA fragments comprising 650 bp of *pstS* and *glmS* and the *parS* region were amplified by PCR using MG1655 genomic DNA as template and the plasmid pFH3228, respectively. After purification, these fragments were assembled via Gibson assembly [61] into EcoRV-digested pNPTS138-R6KT plasmid, resulting in the pNTPS138-R6KT-*parS_ori* plasmid. The resulting plasmid was introduced into *E. coli* MG1655 by conjugative mating using *E. coli* WM3064 as a donor on LB medium containing DAP. Single-crossover integration mutants were selected on LB plates containing kanamycin but lacking DAP. Single colonies were then streaked out on LB plates containing 10% (w/v) sucrose but no NaCl to select for plasmid excision. Kanamycin-sensitive colonies were then checked for targeted insertion by colony PCR and sequencing of the respective PCR fragment. In order to gain strain *E. coli* MG1655-Δ*cadC parS_ori*, the *parS* site of *Y. pestis* was inserted close to the *ori*, at 84.3′ in *E. coli* MG1655-Δ*cadC*, as described above.

In order to gain strain *E. coli* MG1655_P$_{cadBA}$_terminus, the *cadBA* promoter region was inserted at the terminus (33.7′) in *E. coli* MG1655. Construction of this strain was achieved via double homologous recombination using the pNTPS138-R6KT-P$_{cadBA}$_terminus plasmid [27] as described above. Correct colonies were then checked for targeted insertion by colony PCR and sequencing of the respective PCR fragment.

Details of the strains and plasmids used in this study are summarized in Table 3.

***In vivo* fluorescence microscopy.** To analyze search response of mCherry-CadC to its binding site(s), overnight cultures of *E. coli* MG1655 (one CadC binding site close to *ori*, N-P$_{cadBA}$), *E. coli* MG1655ΔP$_{cadBA}$_P$_{cadBA}$_terminus (one CadC binding site at terminus, T-P$_{cadBA}$) and *E. coli* MG1655_P$_{cadBA}$_terminus (two CadC binding sites, N+T-P$_{cadBA}$), each carrying pET-*mCherry-cadC*, were prepared in KE medium pH 7.6 and aerobically cultivated at 37˚C. The overnight cultures were used to inoculate day cultures (OD$_{600}$ of 0.1) in fresh medium at pH 7.6. At an OD$_{600}$ of 0.5, cells were gently centrifuged and resuspended, thereby exposing them to low pH (KE medium pH 5.8 + lysine). Then the cultures were aerobically cultivated at 37˚C and every 1 min after the shift to low pH, 2 μl of the culture was spotted on 1% (w/v) agarose pads (prepared with KE medium pH 5.8 + lysine), placed onto microscope slides and covered with a coverslip. Subsequently, images were taken on a Leica DMi8 inverted microscope equipped with a Leica DFC365 FX camera (Wetzlar, Germany). An excitation wavelength of 546 nm and a 605 nm emission filter with a 75 nm bandwidth was used for mCherry fluorescence with an exposure of 500 ms, gain 5, and 100% intensity. Before shifting the cells to low pH, 2 μl of the cultures in KE medium pH 7.6 were spotted on 1% (w/v) agarose pads (prepared with KE medium pH 7.6) and imaged as a control.

To analyze the spatiotemporal localization of a chromosomal locus, the *parS* site was inserted close to the *ori*. The localization of the *parS* site was visualized via the binding of ParB-yGFP [65]. *E. coli* MG1655 *parS_ori* cells carrying plasmid pFH3228 were cultivated in KE medium pH 7.6 as described above. At an OD$_{600}$ of 0.5, 2 μl of the culture were shifted on 1% (w/v) agarose pads (prepared with KE medium pH 7.6 or pH 5.8 + lysine) and placed onto microscope slides and covered with a coverslip. Subsequently, every 30 s time lapse images of the same cells were taken on a Leica DMi8 inverted microscope equipped with a Leica DFC365 FX camera (Wetzlar, Germany) of the same positions. An excitation wavelength of 485 nm and a 510 nm emission filter with a 75 nm bandwidth were used for ParB-yGFP fluorescence with an exposure of 350 ms, gain 3, and 100% intensity.

**Microscopy image analysis.** To analyze the fluorescence microscopy images for CadC or ParB spots within the cells, we used Oufti [66], an open-source software designed for the analysis of microscopy data for cell segmentation of the phase contrast microscopy images. The resulting cell outlines were used in a custom-written software implemented in Matlab and available on request to detect fluorescent spots. Briefly, a graphical user interface (GUI) was implemented that allows testing the parameters in a test mode before running the actual detection. In detection mode a function *SpotDetection.m* is called, that iterates through all frames and all cells. For each cell, from pixels in the fluorescence microscopy images the intensity of which is above a threshold defined by the parameters and dependent on the mean and variance of the fluorescence signal within the cell the connected components are computed. The components are checked for minimum and maximum size and minimum distance to other spots before being added to the list of spots. For further computations, information on all cells and spots were saved for all frames corresponding to a certain time after receptor activation.

**Interpretation of CadC spot data.** In a custom-written Matlab script the results from the image analysis were used to compute the fraction of cells with spots *v(t)* as a function of time *t* after receptor activation, which upon normalization corresponds to the CDF of the search time distribution. We fit the data to the CDF of a theoretical model using the *curve_fit* function of the scipy module in Python, choosing a trust region reflective algorithm, which is an evolution of the Levenberg-Marquardt method that can handle bounds. This algorithm was developed to solve nonlinear least squares problems and combines the gradient descent method and the Gauss-Newton method. It minimizes the sum of the weighted squares of the errors

between the measured data $y_i$ and the curve-fit function $\hat{y}(t; \boldsymbol{p})$

$$\chi^2(\boldsymbol{p}) = \sum_{i=1}^{m} \left( \frac{y(t_i) - \hat{y}(t_i; \boldsymbol{p})}{\sigma_{y_i}} \right)^2 ,$$

with an independent variable $t$ and a vector of $n$ parameters $\boldsymbol{p}$ and a set of $m$ data points $(t_i, y_i)$. $\sigma_{y_i}$ is the measurement error for measurement $y(t_i)$ and $W_{ij} = \frac{1}{\sigma_{y_i}^2}$ is the weighting matrix. [67]

**Calculation of the ParB diffusion constant.** The results from the image analysis were used to compute trajectories of ParB spots in a custom-written Matlab script by selecting the closest spots in subsequent image frames.

From the trajectories of ParB spots the ensemble-averaged mean square displacement (MSD) was computed as a function of time lag $\tau$: $\mathrm{MSD}(\tau) = \langle (\vec{r}(t) - \vec{r}(t+\tau))^2 \rangle$, where the mean was taken over different spots and $\tau = n30\mathrm{s}$ with $n \in \{1, \ldots, N\}$ and the number of time steps $N$. The Rouse model predicts the MSD in 2D [68]:

$$
\begin{aligned}
\mathrm{MSD}(\tau) &= \sqrt{\frac{16 l_{\mathrm{K}}^2 k_{\mathrm{B}} T}{3\pi\gamma}} \tau^{0.5} \\
&= \sqrt{\frac{16 l_{\mathrm{K}}^2}{3\pi}} D_0^{0.5} \tau^{0.5} ,
\end{aligned}
\tag{15}
$$

with Boltzmann's constant $k_{\mathrm{B}}$, absolute temperature $T$, Kuhn length $l_{\mathrm{K}}$, friction constant $\gamma$ and bead diffusion constant $D_0$, where we have used $D_0 = \frac{k_{\mathrm{B}} T}{\gamma}$ [69]. We fitted the experimentally determined MSD to $\mathrm{MSD}(t) = \Gamma t^{0.5}$ and determined the diffusion constant from $D_0 = \Gamma^2 \frac{3\pi}{16 l_{\mathrm{K}}^2}$.

## Supporting information

**S1 Fig. Dynamics of the target search by CadC.** Fluorescent microscopy images were taken every minute after receptor activation and analyzed for CadC spots for all three *E. coli* strains. The plot shows the fraction of cells with spots $v(t)$ as a function of time $t$ after the medium shift to low pH and lysine.
(PDF)

**S2 Fig. Mean square displacement of ParB spots.** The MSD of ParB spots was calculated by selecting the closest spots in subsequent image frames and calculating the ensemble-averaged mean square displacement as a function of time lag $\tau$. The dashed lines show the fit to $\Gamma \tau^{\alpha}$. For each time lag the mean was taken over 234 to 936 values.
(PDF)

**S3 Fig. Polymer length dependence of the search time.** The mean search time $\tau$ is plotted against the number of DNA beads $N_{\mathrm{DNA}}$. After an initial increase the search time becomes independent of the polymer length. Simulations were run with the realistic parameter set and varying number of DNA beads.
(PDF)

**S1 Table. Fit results.** Results from fitting the experimentally computed CDF to the sequential reversible model with mixed initial condition (N-P$_{cadBA}$) and fixed initial condition (T-P$_{cadBA}$ and N+T-P$_{cadBA}$). The fit parameters $\alpha$, $\beta$ and $c$ were used to compute the mean first passage time and the variance with uncertainties obtained from error propagation using the full covariance matrix.
(PDF)

**S2 Table. Covariance matrix.** Covariance matrix of the parameters $\alpha$, $\beta$ and $c$ from fitting the experimentally computed CDF to the sequential reversible model with mixed initial condition (N-$P_{cadBA}$) and fixed initial condition (T-$P_{cadBA}$ and N+T-$P_{cadBA}$).
(PDF)

**S1 Text. Supplementary text.** Derivation of the cumulative distribution function and the first two moments for the sequential reversible process.
(PDF)

## Acknowledgments

We thank Flemming G. Hansen, Technical University of Denmark, for providing the plasmids pFHC2973 and pFH3228.

## Author Contributions

**Conceptualization:** Kirsten Jung, Ulrich Gerland.

**Data curation:** Linda Martini, Sophie Brameyer, Elisabeth Hoyer.

**Formal analysis:** Linda Martini.

**Funding acquisition:** Kirsten Jung, Ulrich Gerland.

**Investigation:** Linda Martini, Sophie Brameyer, Elisabeth Hoyer.

**Methodology:** Linda Martini, Sophie Brameyer, Elisabeth Hoyer, Kirsten Jung, Ulrich Gerland.

**Project administration:** Kirsten Jung, Ulrich Gerland.

**Resources:** Kirsten Jung, Ulrich Gerland.

**Software:** Linda Martini.

**Supervision:** Kirsten Jung, Ulrich Gerland.

**Validation:** Linda Martini, Sophie Brameyer, Elisabeth Hoyer, Kirsten Jung, Ulrich Gerland.

**Visualization:** Linda Martini, Sophie Brameyer.

**Writing – original draft:** Linda Martini, Sophie Brameyer, Ulrich Gerland.

**Writing – review & editing:** Linda Martini, Sophie Brameyer, Elisabeth Hoyer, Kirsten Jung, Ulrich Gerland.

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
