## [Decision Letter · Decision Letter 0]

8 Sep 2020

Dear Dr. Gerland,

Thank you very much for submitting your manuscript "Dynamics of chromosomal target search by a membrane-integrated one-component receptor" for consideration at PLOS Computational Biology. As with all papers reviewed by the journal, your manuscript was reviewed by members of the editorial board and by several independent reviewers. The reviewers appreciated the attention to an important topic. Based on the reviews, we are likely to accept this manuscript for publication, providing that you modify the manuscript according to the review recommendations.

Thank you for your submission. Based on the three reviews received, your paper presents a valuable contribution and requires minor revisions to address several points raised by the reviewers. These points attempt to clarify features and assumptions of the model, as well as context of your conclusions.

Sincerely,

Tamar Schlick

Associate Editor

PLOS Computational Biology

Daniel Beard

Deputy Editor

PLOS Computational Biology

[LINK]

Thank you for your submission. Based on the three reviews received, your paper presents a valuable contribution and requires minor revisions to address several points raised by the reviewers. These points attempt to clarify features and assumptions of the model, as well as context of your conclusions.

Reviewer's Responses to Questions

**Comments to the Authors:**

Reviewer #1: Review uploaded as an attachment

Reviewer #2: Review is uploaded as an attachment.

Reviewer #3: Understanding the machinery which cells use to respond to signals from their environment is one of the fundamental goals of cell biology. While the most studied sensing systems use both sensors on the cell surface and molecules which transmit that signal to the DNA to change gene expression, in this paper the authors study a system in which one molecule does both tasks. This poses an interesting question: if the molecule doesn’t leave the membrane, how does it find the right spot on the hairball of DNA inside the cell?

First, the authors measure the time for the sensor, CadC, to bind the DNA in many cells. They then use a three state model to contextualize the measurements and find mean search times. The authors then build a 3D model which incorporates spatiotemporal dynamics of both DNA and CadC. They do experiments to estimate values for the parameters on which this model depends and show that the model predicts search times close to the measured value, a significant achievement. They then show the model also agrees with a well chosen perturbation experiment, another significant achievement. Taken together, these results suggest that DNA and sensor mobility together allow search times on the same order of magnitude of more conventional sensing systems in which the sensor and signal transmitter are separate molecules, an interesting result.

The study is well executed and its results are of significant biological interest. While I am not an expert in bacteria or signal transduction, it is clear that the experimental and modeling efforts are strong individually. Furthermore, because they are tightly integrated, their impact is even greater than the sum of these parts. It is an example of skills and tools coming together from different realms to gain understanding that could not be achieved by either experiment or modeling alone. The authors have done a wonderful job gathering data which addresses the model parameters directly, and making a model which addresses directly what is measurable. Additionally, the paper is well written and very clear in its goals and execution.

Addressing the following point is critical for the conclusions of the paper:

I. In simulation, the authors assume there is one CadC dimer on the cell membrane. However, they state that either 1-3 or 3-5 molecules are present on the membrane. A priori, having 2 or more dimers vs. 1 could impact the search time significantly. The authors should address this point.

The following suggestions would help improve the clarity of the paper:

1. ori should be defined

2. The layout of figure 4 distracts from its main points, and could be improved by some small changes

a. All data should be represented with errorbars to contextualize wiggles (Does NDNA=125 really have a longer search time than the surrounding values?)

b. k1D is included in 4B and its legend, but elsewhere the text says sliding was not included in the simulation. Suggest removing is, as well as associated text in “Biophysical model of the target search process”

c. 4C is a figure which supports the choice of a parameter value, but does not yield understanding about the biology the authors are addressing. Suggest moving it to the supplement.

d. The prominence of the dashed line in 4D detracts from the plotted data. Suggest stopping the y axis at 10^-1. An annotation including the value for the relevant experiment would better help the reader contextualize this nice result.

e. Adding an annotation with the experimental mean time to 4E and 4F would help readers understand that the model closely matches the experiment at the measured parameters, a significant achievement which deserves to be highlighted.

f. The dashed line should likely not extend across the axis break in 4F

3. The authors question if DNA or CadC mobility is “more crucial” or “contributes more”. This language is not well defined and detracts from otherwise superb writing. Suggest authors phrase this result in the context of either which timescale limits the search, or which rate the search is more sensitive to changes in.

4. The naming and value of NDNA are inconsistent (in 4C and text NDNA=100, while in table 2 NDNA=10um)

**Have all data underlying the figures and results presented in the manuscript been provided?**

Reviewer #1: None

Reviewer #2: Yes

Reviewer #3: Yes

PLOS authors have the option to publish the peer review history of their article (what does this mean?). If published, this will include your full peer review and any attached files.

Reviewer #1: **Yes: **Naveed Aslam

Reviewer #2: **Yes: **James D Brunner

Reviewer #3: **Yes: **Matthew Bovyn
---

## [Decision Letter · Decision Letter 1]

16 Dec 2020

Dear Dr. Gerland,

Thank you very much for submitting your manuscript "Dynamics of chromosomal target search by a membrane-integrated one-component receptor" for consideration at PLOS Computational Biology. As with all papers reviewed by the journal, your manuscript was reviewed by members of the editorial board and by several independent reviewers. The reviewers appreciated the attention to an important topic. Based on the reviews, we are likely to accept this manuscript for publication, providing that you modify the manuscript according to the review recommendations.

Please address remaining comment of Reviewer 2.

Sincerely,

Tamar Schlick

Associate Editor

PLOS Computational Biology

Daniel Beard

Deputy Editor

PLOS Computational Biology

[LINK]

Please address remaining comment of Reviewer 2.

Reviewer's Responses to Questions

**Comments to the Authors:**

Reviewer #1: Authors have done a great job in addressing all the issues raised earlier and they have modified the manuscript to further enhance its impact. I recommend this for publication.

Reviewer #2: The authors use Monte Carlo simulation to model a proposed mechanism for a one-component receptor & transcriptional activator in bacteria, using E. Coli CadC as an example system. Comparison with experimental data makes a convincing case that this mechanism exhibits plausible time-scales, an important necessary condition for the mechanism's accuracy. The paper explains the mechanism and model well. Overall, I think this is a very good use of modeling and simulation.

My only comment is that the use of the two-state stochastic system to fit parameters and thereby compute a mean first passage time still seems unnecessary. The fitted parameters seem to be used just to compute the mean, but this can be computed directly from the observed CDF. There are likely other advantages to having a functional form of the distribution that will be used in future work, and the authors could add some comment as to what these are.

**Have all data underlying the figures and results presented in the manuscript been provided?**

Reviewer #1: Yes

Reviewer #2: Yes

PLOS authors have the option to publish the peer review history of their article (what does this mean?). If published, this will include your full peer review and any attached files.

Reviewer #1: **Yes: **Naveed Aslam

Reviewer #2: **Yes: **James D. Brunner
---

## [Editor Report · Decision Letter 2]

7 Jan 2021

Dear Dr. Gerland,

We are pleased to inform you that your manuscript 'Dynamics of chromosomal target search by a membrane-integrated one-component receptor' has been provisionally accepted for publication in PLOS Computational Biology.

Best regards,

Tamar Schlick

Associate Editor

PLOS Computational Biology

Daniel Beard

Deputy Editor

PLOS Computational Biology

---

## [Editor Report · Acceptance letter]

30 Jan 2021

PCOMPBIOL-D-20-01259R2 

Dynamics of chromosomal target search by a membrane-integrated one-component receptor

Dear Dr Gerland,

I am pleased to inform you that your manuscript has been formally accepted for publication in PLOS Computational Biology. Your manuscript is now with our production department and you will be notified of the publication date in due course.

With kind regards,

Alice Ellingham
